

# Benthic Alkalinity fluxes from coastal sediments of the Baltic and North Seas: Comparing approaches and identifying knowledge gaps

Bryce Van Dam[1,*], Nele Lehmann[1,4,7] Mary A. Zeller[3], Andreas Neumann[1], Daniel Pröfrock[2], Marko Lipka[3], Helmuth Thomas[1,7], Michael E. Böttcher[3,5,6]

[1] Helmholtz-Zentrum Hereon, Institute of Carbon Cycles, Geesthacht, Germany
[2] Helmholtz-Zentrum Hereon, Institute of Coastal Environmental Chemistry, Geesthacht, Germany
[3] Leibniz Institute for Baltic Sea Research (IOW), Warnemünde, Germany
[4] Alfred Wegener Institute Helmholtz Centre for Polar and Marine Research, Potsdam, Germany
[5] Marine Geochemistry, University of Greifswald, FRG
[6] Interdisciplinary Faculty, University of Rostock, FRG
[7] University of Oldenburg, Oldenburg, Germany

*Correspondence to*: Bryce Van Dam (Bryce.Dam@hereon.de)

**Abstract.** Benthic alkalinity production is often suggested as a major driver of net carbon sequestration in continental shelf ecosystems. However, information and direct measurements of benthic alkalinity fluxes are limited and are especially challenging when biological and dynamic physical forcing causes surficial sediments to be vigorously irrigated. To address this shortcoming, we quantified net sediment-water exchange of alkalinity using a suite of complementary methods, including 1) $^{224}$Ra budgeting, 2) incubations with $^{224}$Ra and Bromide as tracers, and 3) numerical modelling of porewater profiles. We choose a set of sites in the shallow southern North Sea and western Baltic Sea, allowing us to incorporate frequently occurring sediment classes ranging from coarse sands to muds, and sediment-water interfaces ranging from biologically irrigated and advective to diffusive into the investigations. Sediment-water irrigation rates in the southern North Sea were approximately twice as high as previously estimated for the region, in part due to measured porewater $^{224}$Ra activities higher than previously assumed. Net alkalinity fluxes in the Baltic Sea were relatively low, ranging from an uptake of -35 μmol m$^{-2}$ hr$^{-1}$ to a release of 53 μmol m$^{-2}$ hr$^{-1}$, and in the North Sea from 1 to 33.6 μmol m$^{-2}$ hr$^{-1}$. Lower than expected apparent nitrate consumption (potential denitrification), across all sites, is one explanation for our small measured net alkalinity fluxes. Carbonate mineral precipitation and sulfide re-oxidation also appear to play important roles shaping net sediment-water fluxes in the North Sea and Baltic Sea sites, respectively.

## 1 Introduction

Continental shelf systems are considered to act as sinks for atmospheric $CO_2$ (Borges et al., 2005; Laruelle et al 2010; 2018; Rutherford et al., 2021), with an important but currently uncertain role in the global carbon budget (Lacroix et al., 2021). Current trends in shelf $CO_2$ uptake are highly variable (Laruelle et al., 2018), in response to a variety of factors, including: regional trajectories in primary production, acid-base buffering (Thomas et al., 2007), the underlying geology, mixing with





riverine and open-ocean waters, and wind patterns (Meyer et al., 2018). Where water depths are relatively shallow, there is an
increased potential for benthic-pelagic coupling to affect the net shelf $CO_2$ uptake. In regions like the shallow southern North
Sea, persistent wind and tidal-driven mixing strongly impacts both the benthic and pelagic ecosystems. This mixing ventilates
surface sediments, supplying oxidants for the respiration of organic matter and flushing out the reduced products, thereby
affecting net sediment-water exchange of dissolved substances like inorganic carbon (DIC) and total alkalinity (TA). Aerobic
remineralization is a key DIC source in physically or biologically ventilated surface sediments (Neumann et al., 2021;
Rassmann et al., 2020), followed by microbial sulfate reduction (MSR; Al-Raei et al., 2009; Werner et al., 1999, 2001). These
DIC sources may be further enhanced to varying extents by anaerobic processes like denitrification, as well as iron and
manganese reduction, which, like MSR also produce TA in addition to DIC (e.g., Zeebe & Wolf-Galdrow, 2001). While intense
biological reworking limits net organic carbon storage in North Sea sediments (Diesing et al., 2020; de Haas et al., 1997), net
carbon uptake from the atmosphere may be possible through a combination of the shelf $CO_2$ pump (Thomas et al., 2004), as
facilitated by large oceanic inflows (Lacroix et al., 2021), as well as a long-term increase in the DIC inventory driven by
combined biological and anthropogenic $CO_2$ (Clargo et al., 2015).

        Additionally, there is a suite of geochemical processes which may also affect net shelf $CO_2$ uptake. For example, the
precipitation and dissolution of calcium carbonate ($CaCO_3$) and silicate minerals, pyrite formation and oxidation, dissolution
and formation of silicate minerals, and reverse weathering may substantially affect net sediment-water fluxes of DIC and TA,
and, hence, the carbonate chemistry of the overlying water column (Berelson et al., 2019; Hagens et al., 2015; Pätsch et al.,
2018; Rassmann et al., 2020; Thomas et al., 2009; Winde et al., 2017; Schwichtenberg et al., 2020). While not a direct result
of biological metabolism, the kinetics of these abiotic processes are determined by the biologically mediated redox setting
impacting the physico-chemical saturation states (Morse and Mackenzie, 1990). Therefore, a strong interplay exists in shelf
sediments, between biogeochemical cycling of macro and minor nutrients, and benthic (bio)geochemical solid-solution
interactions, all of which is further mediated by wind- and tidal-driven physical mixing.





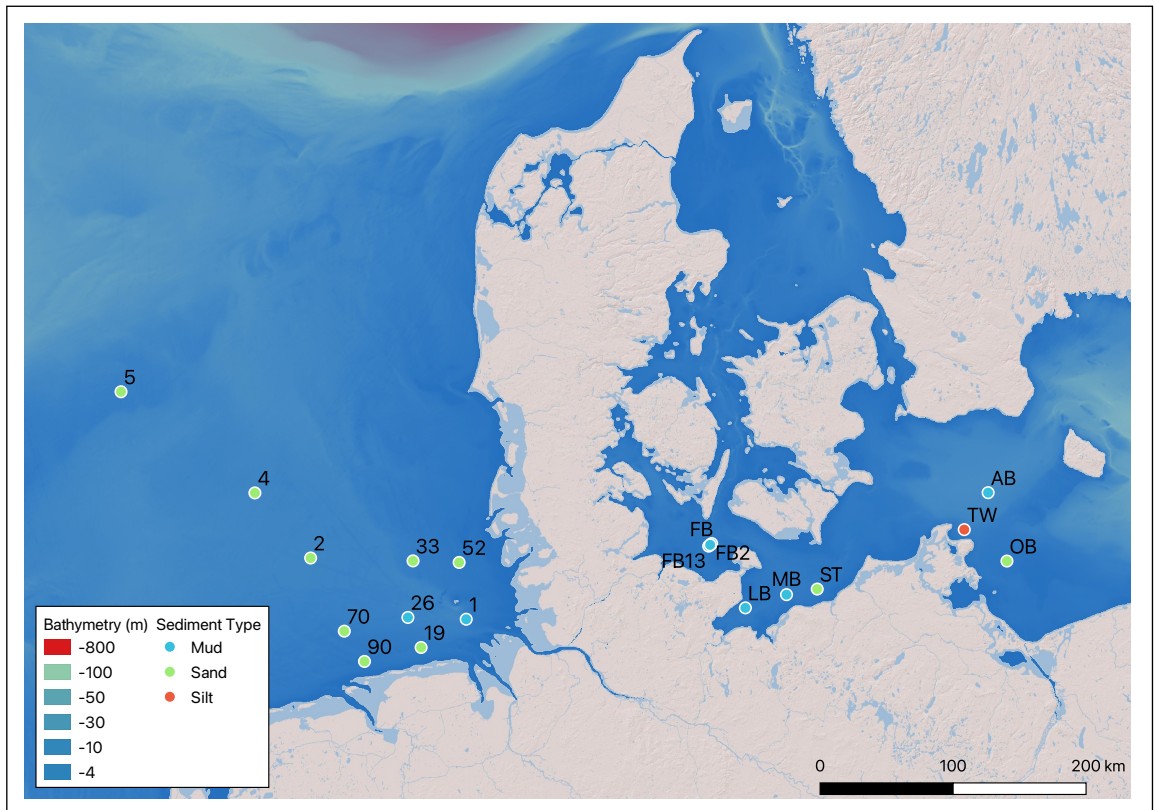

**Figure 1**. Map of sampled stations, with points colored by sediment type.

Substantial prior work in the coastal North Sea shows that these sediment (bio)geochemical processes exert a strong cumulative effect on water-column carbonate chemistry (Brenner et al., 2016; Burt et al., 2014) and ultimate $CO_2$ exchange with the atmosphere (Thomas et al., 2009). Recent modelling and experimental efforts point to benthic denitrification of riverine nitrate as a dominant TA source to the southern North Sea, but also imply a substantial role of benthic carbonate mineral dissolution (Winde et al., 2014; Pätsch et al., 2018; Burt et al., 2016; Schwichtenberg et al., 2020). In addition to these processes, metal (Fe and Mn) reduction can also be an important TA source, especially in depositional centers with finer-grained sediment and greater organic carbon loads (Lenstra et al., 2019; Reithmaier et al., 2021). However, rapid sediment accumulation may decouple net Fe reduction from sediment-water TA fluxes (Rassmann et al., 2020). Hence, more investigations of the benthic (bio)geochemical processes in relation to benthic DIC/TA fluxes is clearly warranted.

The Baltic and North Seas form a continuum of water and material transport (Gustafsson 1997; Kuliński et al., 2022; Maar et al., 2011), deeply linking the carbon cycles of these two ocean basins. While the nearshore regions of the southern North and western Baltic Seas are located at a similar latitude, have a similar water depth, and contain similar sediment types, they experience very different physical forcing. While tidal forcing dominates in the southern North Sea, the western Baltic is instead characterized by negligible tidal forcing, estuarine mixing and associated salinity variability. For example, large tidal



forcing combined with coarse-grained sediments in the southern North Sea, could promote advective over diffusive fluxes, and increase the oxygen penetration depth (Werner et al., 1999, 2003; Billerbeck et al. 2006; Al-Raei et al., 2009). Bioturbation and bioirrigation are also especially important in the coastal North Sea and can explain how sediment-water fluxes of oxygen and other elements vary across season and sediment type (Lipka et al., 2018; Gogina et al., 2018; Neumann et al., 2021; Bratek et al., 2020). In contrast, limited tidal forcing combined with finer-grained sediments in the western Baltic Sea could promote stronger biogeochemical zonation and a relatively greater importance of anaerobic over aerobic processes (Böttcher et al., 2000). While we know that these factors shape benthic biogeochemical processing, sediment-water fluxes in these shallow coastal environments are still poorly-constrained, limiting our understanding of their role in the regional carbon cycle (Kuliński et al., 2022).

In this study, we present new results quantifying sediment-water DIC and TA fluxes in the southern North Sea, considering a variety of methods, including Ra budgets and core incubations. This dataset is augmented by archived porewater data from the Baltic Sea and some additional sites in the North Sea, in order to further understand the differential impacts of physical and biogeochemical forcing in these two nearby basins. Benthic TA, DIC, and element fluxes were modelled from these porewater profiles, aimed at predicting net sediment-water fluxes of DIC and TA across a broad range in sediment types. The relationships of especially $SO_4^{2-}$, DIC, TA, Fe, and $PO_4^{3-}$ were utilized as a first approach on the potential role of secondary (bio)geochemical processes described above, which are so far only poorly represented in regional carbon budgets (Pätsch et al., 2018; Schwichtenberg et al., 2020), in the context of the coupled physical and biological drivers of net benthic-pelagic coupling. Each applied method has its own advantages and disadvantages to emphasize advection, diffusion, or bioirrigation to varying degrees. The differences between the approaches are partially related to methodological limits, but may also indicate real differences between the consequences of physical and biological drivers on biogeochemical processes such as net $CO_2$ uptake and, therefore, require further investigations.

## 2 Methods

### 2.1 Dataset description

Samples collected for this study come from a set of 19 stations in the southern North Sea (10 sites) and the western Baltic Sea (9 sites) that were visited during cruises between 2015 and 2020 (Figure 1). Here we present results from four cruises where sampling and experimental approaches are comparable; these were HE541 (Sept 2019) and MSM50a (Jan 2016) in the North Sea, and EMB111 (Aug-Sept 2015) and EMB238 (May 2020) in the Baltic Sea. Parts of the experimental results from MSM50 and EMB111 have previously been published by Lipka et al. (2018) and Gogina et al. (2018), and these sites were chosen to be complementary in terms of water depth, distance from shore, and sediment type. Only results from HE541 in the southern North Sea are used for the Ra budget and core incubations, while porewater profile modelling and constituent ratio analyses are given for all sites. The water depth at these sites is shallow at 23.6 +/- 8.5 m (mean +/- S.D.), with a maximum



of 46.0 m (Site AB) and minimum of 11.2 m (Site 52). Of the 19 stations considered here, 10 are classified as sandy sediments,
8 as muds, and one as silt sediment (Lipka, 2017; Lipka et al., 2018).

At each site, a multicoring device was used to collect 10 cm diameter cores which contained approximately 20 cm of sediment. Sediment porewaters were extracted within 6 hrs after core collection using Rhizons (nominal pore size of 0.15 µm) at 1-2 cm intervals for the first 10 cm, and with a typical interval of 5 cm below that. Table S1 in the Supporting Information lists the preservation techniques and measurement methods for each analyzed parameter.

**2.2 Lab Analyses**

The samples taken during the HE541 cruise were analyzed according to the following protocols: TA was determined at IOW via a small volume, two-point potentiometric titration according to van den Berg and Rogers, 1987: The sample (400 µL) was transferred into a vial with a pre-set volume of acid on board, assuming that all weak acids in solution would be protonated (pH < 3). The sample was cooled at 4 °C until further analysis. Back in the lab, the sample was allowed to equilibrate
to room temperature and the first potentiometric point was measured using a pH meter (SevenMulti, Mettler-Toledo, Gießen, Germany). After the addition of HCl (75 µL, 0.1 M), the second point was recorded (pH ≈ 2). Taking these two points, the slope of the electrode response was determined (van den Berg and Rogers, 1987). The measurements were calibrated against certified reference materials (CRMs), from batch 142 provided by Prof Andrew Dickson (Scripps Institution of Oceanography, San Diego, USA).

DIC and $\delta^{13}$C-DIC were also analyzed at IOW with a continuous-flow isotope-ratio-monitoring mass spectrometry (CF-irmMS) using a gas mass spectrometer (Finnigan MAT 253, Thermo Fisher Scientific, Waltham, USA) coupled to a gas bench (GasBench II, Thermo Fisher Scientific, Waltham, USA) via a continuous flow interface (ConFlo IV, Thermo Fisher Scientific, Waltham, USA) (Winde et al., 2014a). A calibration of the instrument was performed against CRMs provided by Prof Andrew Dickson (Scripps Institution of Oceanography, San Diego, USA). Sulfide concentrations were measured at IOW
using the methylene blue technique (Cline, 1969) on a Specord 40 a spectrophotometer (Analytik Jena, Germany).

Determination of elemental concentrations (Ca, Fe, Mn, P, S) was performed at Hereon using an ICP-MS/MS (Agilent 8800, Agilent Technologies, Tokyo, Japan) coupled to an ESI SC-4 DX FAST autosampler (Elemental Scientific, Omaha, Nebraska, USA) equipped with a discrete sampling system with a loop volume of 1.5 mL (Zimmermann et al., 2020). Measurements were validated with a seawater standard (S = 35, IAPSO Standard Seawater, Ocean Scientific International Ltd,
Hampshire, United Kingdom) with the addition of Fe and Mn (c = 1 mg/L). Calibrations were prepared from either single or multi element solutions traceable to NIST CRMS. Nutrient concentrations were determined using an automated continuous flow system (AA3, Seal Analytical, Norderstedt, Germany) and standard colorimetric techniques (Graßhoff and Almgren, 1983). Dissolved metals, S, P, and Si results from the Baltic Sea samples were measured at IOW via ICP-OES at IOW as described by Winde et al. (2014a) and Lipka et al. (2018).





## 2.3 Sediment-water flux determinations


In this study, we applied three independent methods to estimate sediment-water fluxes of major, minor, and trace elements. This is a conservative approach, because each individual method emphasizes advective or diffusive processes to a different degree, ensuring that our calculated sediment-water fluxes span a complete range in possible rates.

First, a water-column $^{224}Ra$ decay balance, used to derive porewater irrigation rates, which are in turn used to
parameterize bulk sediment-water fluxes based on measured concentrations in the upper porewaters. The second approach is similar to the first, but with irrigation rates derived from ship-board core incubation experiments, with both bromide and $^{224}Ra$ employed as tracers. We also used PROFILE models to assess sediment-water fluxes for all sites, which uses the shape of porewater profiles to balance internal production and consumption with a vertical diffusive transport (Berg et al., 1998).

## 2.4 $^{224}Ra$ decay balance


Samples for $^{224}Ra$ were collected exclusively during the HE541 cruise in the southern North Sea, following the methods of Moore et al. (2011) and Burt et al., 2016 ($^{224}Ra$ determinations were not made during the other cruises). Briefly, approximately 100 L of water from the ship's seawater line was passed through a cascade of 10 and 1 μm filters, then pumped slowly (1 L/min) through a cartridge containing manganese oxide-coated fibers which quantitatively adsorb Ra (confirmed with efficiency samples). After rinsing and drying, samples were counted first within 24 hours of collection on board the ship
with a Radium Delayed Coincidence Counting (RaDeCC) system (Moore and Arnold 1996). A second count was executed after 7-14 days, allowing us to apply the error propagation technique of Burt et al., 2016 to derive uncertainty statistics for the final $^{224}Ra$ activities given in section 3.1. In addition to the bulk seawater samples, porewaters were also collected from all sites during the HE541 cruise using Rhizons, with water from all core depths combined into a single sample (~100-200 mL) that was treated and counted on a RaDeCC as described above.


Because $^{224}Ra$ is not a gas and has a relatively short half-life (3.7 days), its activity in surface waters is a relatively simple balance between advective-diffusive inputs from the sediments (where it decays from its longer-lived parent isotope $^{228}Th$), lateral exchange in surface waters, and radioactive decay (Garcia-Orellana et al., 2021). We assume that lateral exchange is a small term in this budget, allowing us to calculate the benthic advective-diffusive input term (net irrigation) as the balance between radioactive decay and the measured surface water activity. These irrigation rates derived from the $^{224}Ra$
decay balance (mean = 81 L m$^{-2}$ d$^{-1}$) were then used to calculate net sediment-water fluxes by multiplying by the average concentration difference between the pore water and the overlying water concentration. For "sandy" sites, we applied the average porewater concentration of each parameter from the upper 5 cm in each core, while a single concentration from the top sediment layer of each core was used for the "muddy" and "silty" sites. Sign convention is that positive fluxes indicate a flux out of the sediment and vice versa.





## 2.5 Ship-board incubations

Benthic fluxes were measured on intact sediment cores from Multicorer by means of whole core batch incubations. Typically, 3 to 4 intact sediment cores in transparent plastic liners (PMMA, 10 cm inner diameter, 60 cm length) per station were selected, which had no visible perturbations such as cracks, voids, or injured animals. The sediment cores were typically 15 - 30 cm in length. The incubation was enclosed with a gas-tight lid that was adjusted to a resulting supernatant height of 15 cm (approx. 1 L volume). The water column was constantly stirred by horizontal propellers. The stirring intensity was adjusted to the highest intensity that did not result in sediment resuspension to ensure vigorous mixing of the supernatant and prevent suspended particles from settling. Incubations were executed aboard the research vessel in a temperature-controlled lab set to in-situ temperature. Primary production was excluded by wrapping the cores in aluminum foil. At the beginning of each incubation, a NaBr solution was injected into the supernatant, which resulted in a final Br⁻ concentration of 1.6 mmol / L. During incubation, oxygen was monitored continuously to assess the progress, and the incubation was terminated when the oxygen saturation dropped below approximately 80 % (typically after 12 to 24 h). Water samples for Bromide analysis were drawn from the core supernatant in 3 – 6 h intervals (typically 10 time steps) by means of syringes connected to PVC-tubing. Water samples were then filtered through 0.45 µm syringe filters and stored frozen until analysis in the land-based laboratory. After incubations, additional samples form the supernatant were taken for measurements of Radium and DIC. The incubation method is described in detail in Neumann et al. (2021).

## 2.6 PROFILE Modelling

We first employed the one-dimensional numerical model tool, PROFILE, to derive sediment-water fluxes and internal rates of production and consumption for most measured constituents (Berg et al., 1998). PROFILE builds a best-fit model from measured porewater profiles, assuming they are representative of steady state conditions. This model is then split into several equidistant zones (typically 5 to 10 discrete intervals), each of which is fit with a specific net production rate. We then set a first boundary condition as the calculated diffusive flux out the bottom of the model domain, using the concentration gradient in the two deepest porewater samples, as well as the sediment porosity and calculated the molecular diffusion coefficient (Table 1). The final sediment-water flux is then determined through conservation of mass between net internal production and the bottom diffusive flux. Further parameters needed to inform PROFILE include: sediment porosity ($\varphi$), biodiffusivity, irrigation, diffusivity, statistical terms, and others given in Table 1.

**Table 1.** Parameterization of PROFILE model

|  |  | Sand | Silt | Mud |
|---|---|---|---|---|
| **Max deviation when accepting value (%)** |  | 0.001 | 0.001 | 0.001 |
| **Level of significance for F-statistic** |  | 0.2 | 0.2 | 0.2 |
| **Biodiffusivity (cm² hr⁻¹)** | < 5 cm | 0.36 | 0.18 | 0.18 |
|  | 5-10 cm | 0.18 | 0 | 0 |
|  | >10 cm | 0 | 0 | 0 |



| Irrigation coefficient (hr$^{-1}$) | < 5 cm | 0.11 | 0.0011 | 0.0011 |
|---|---|---|---|---|
| | 5-10 cm | 0.011 | 0 | 0 |
| | >10 cm | 0 | 0 | 0 |
| **Porosity ($\varphi$)** | | 0.4 | 0.6 | 0.8 |
| **Coefficient of Molecular Diffusivity (cm$^2$ hr$^{-1}$)** | | Calculated in R: 'marelac' as $f$(Sal, Temp, Pres) | | |
| **Sediment Diffusivity (cm$^2$ hr$^{-1}$)** | | $D_s = {D}/{(1 + 3[1 - \varphi])}$ | | |
| **Boundary Conditions** | | Concentration and diffusive flux at the bottom | | |

## 3 Results and Discussion

### 3.1 North Sea Irrigation rates (decay balance and incubation)

Surface water $^{224}$Ra activities were consistent throughout the southern North Sea, ranging from 248 ± 11.6 dpm m$^{-3}$

(Site 33) to 384 ± 21.3 dpm m$^{-3}$ (Site 90). This is approximately twice as high as found in previous studies in the region (Burt et al 2014; 2016), but similar to activities for the Wadden Sea (Moore et al., 2011). Our observation of high surface water $^{224}$Ra is consistent with early fall wind-driven vertical mixing causing an enhanced exchange between porewater high in $^{224}$Ra and the surface water. The wind-driven water column mixing also had the effect of breaking down any vertical stratification that may have developed during the summer (not shown). Our porewater $^{224}$Ra activities (8.4 ± 0.89 to 44 ± 3.0 dpm L$^{-1}$; average

of 19.9 dpm L$^{-1}$ (or 19,946 dpm m$^{-3}$)) were also approximately double compared to previous measurements in the German Bight (Burt et al., 2014), and the Wadden Sea (Moore et al., 2011). These high $^{224}$Ra activities yield calculated irrigation rates (mean = 81 L m$^{-2}$ d$^{-1}$; red bars in Figure 2) that were about twice the levels found by Burt et al., (2014). Solute fluxes for these sites are derived using PROFILE modelling, as discussed later. The high variability in porewater $^{224}$Ra activity (8.4 to 44 dpm L$^{-1}$) also shows that care should be taken when deciding on endmember activities for $^{224}$Ra-based budgets aimed at assessing

porewater irrigation rates (Cook et al., 2018; Garcia-Orellana et al., 2021).

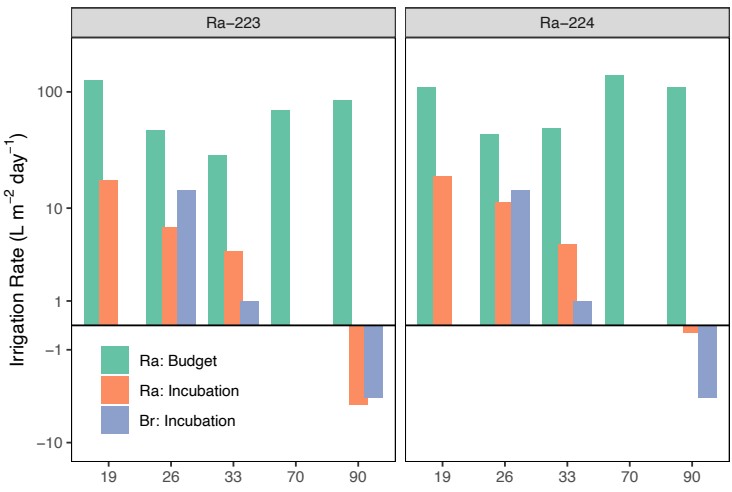



**Figure 2.** Porewater irrigation rates (L m$^{-2}$ d$^{-1}$) derived from Ra budget (green) and incubations with Ra (orange) or Bromide (blue) as tracers.

Irrigation rates derived from Br and $^{224}$Ra measurements during the ship-board incubation experiments (mean = 10.5 L m$^{-2}$ d$^{-1}$; blue bars) were a factor of 8 lower than rates derived from the $^{224}$Ra decay balance. This is consistent with prior studies in the southern North Sea, where incubation-based fluxes were 2-3 times lower than decay-balance estimates (Burt et al 2014). This difference is likely due to the fact that the decay balance implicitly represents the combined diffusive, advective, and bio-irrigative processes that exist in the environment, which are incompletely captured in whole-core incubations. We must stress
here that the Ra decay-balance does not necessarily represent a better estimate of the real irrigation rate, as many issues also exist with this method especially related to porewater endmember determination and the assumption of steady state (Cook et al., 2018; Garcia-Orellana et al., 2021). Rather, these irrigation rates and solute fluxes presented below, simply represent a most likely range in "real-world" values. The difference among approaches can also illustrate the varying importance of physical and bio-mediated forcing of net sediment-water exchange.

**3.2 Net TA Fluxes**

**3.2.1 North Sea TA fluxes from irrigation, incubation, and PROFILE modelling**

    Using the range in irrigation rates shown above, we calculated net sediment-water fluxes for a variety of solutes using the measured surface-pore water concentration difference (by convention, positive is out of the sediment). Average TA fluxes are tabulated in table 2. At only two sites in the North Sea was porewater TA enriched enough to reliably calculate net fluxes
using the incubation or budget approaches, given the relatively low precision of the small-volume TA titration method applied here. However, at these two sites (19 and 26), our TA fluxes (Figure 3) were 1-2 orders of magnitude lower than previous estimates for this region (Brenner et al., 2016; Burt et al., 2014; Voynova et al., 2019). Excluding the PROFILE-modelled TA fluxes for the North Sea (which had very low R$^2$), net TA fluxes in the North Sea were relatively low, ranging from 1 to 6.3 µmol m$^{-2}$ hr$^{-1}$ across all sites and sediment types (Table 2).


**Table 2.** Comparison of average TA fluxes (µmol m$^{-2}$ hr$^{-1}$) between methods in the North and Baltic Seas, separated by sediment type.

| Basin | Method | Mud | Sand | Silt |
|---|---|---|---|---|
| **Baltic Sea** | PROFILE | 53 ± 95.3 (n=6) | -4.3 ± 43.5 (n=2) | -35 (n=1) |
| **North Sea** | PROFILE | 1 ± 2.1 (n=3) | 33.6 ± 42.3 (n=5) | - |
| **North Sea** | Ra Incubation | 1.5 ± 3.5 (n=2) | 1.1 ± 1.5 (n=2) | - |
| **North Sea** | Ra Budget | 5.6 ± 13.4 (n=2) | 6.3 ± 8.8 (n=2) | - |
| **North Sea** | Br Incubation | 1.8 ± 4.4 (n=2) | - | - |



Burt et al., (2014) applied a similar Ra-based approach to estimate benthic fluxes but used a porewater-surface water
TA difference of 0.7 mM, far greater than what we observed. For example, our porewater TA measurements at sites 19 and 26
were on average 2.23 and 2.26 mM, only 2 - 4 μM (0.002 – 0.004 mM) greater than bottom water TA. As a result, the inferred
sediment-water TA fluxes in Burt et al., 2014 were substantially larger than ours, at 196-921 μmol m$^{-2}$ hr$^{-1}$. Likewise, Voynova
et al., 2019 estimated net spring-summer sediment-water TA fluxes in the southern North Sea to be more than two orders of
magnitude higher, at 488-1117 μmol m$^{-2}$ hr$^{-1}$ (Table 1 in Voynova et al., 2019). More in line with our findings are the results
of Brenner et al., 2016, who directly measured TA fluxes in ex-situ incubations of 237.5 - 275 μmol m$^{-2}$ hr$^{-1}$ (Southern North
Sea, 2011 and 2012), and a modelling simulation placing mean TA fluxes at 83 μmol m$^{-2}$ hr$^{-1}$ (Pätsch et al., 2018).

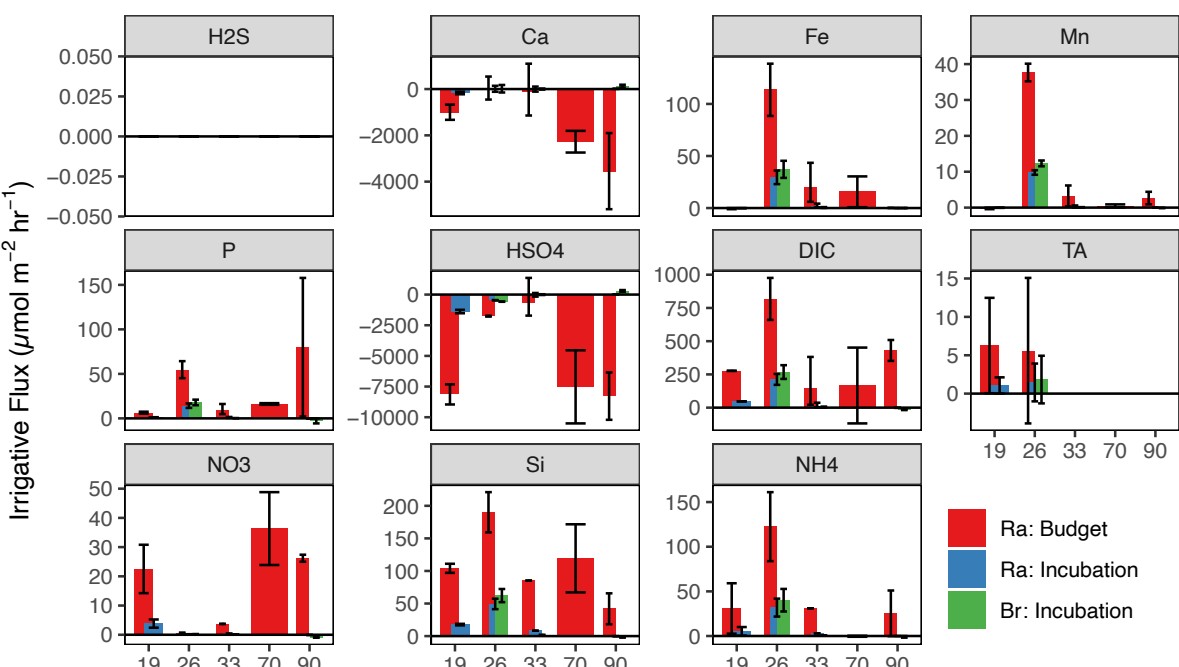

**Figure 3**. Average sediment-water fluxes (μmol m$^{-2}$ hr$^{-1}$) derived from Ra or Br incubation or Ra-decay from the North Sea
sites only. In this figure, the error bars represent minimum and maximum fluxes calculated from replicate pore-water cores
taken at each site.

### 3.2.2 Baltic Sea TA fluxes from PROFILE modelling

In the Baltic Sea muddy sites (AB, LB, MB), TA concentration in surface sediments (0-10 cm depth) exceeded surface
water TA by 1-10 mM. This excess TA caused modelled sediment-water TA fluxes in the Baltic Sea muddy sites as high as
159 μmol m$^{-2}$ hr$^{-1}$ (site LB, Figure 4). Down-core trends in TA at these sites are well within measurement precision, allowing
us to discuss potential sources of this TA and implications in section 3.5. In contrast with the relatively large excess TA in
surface sediments at the Baltic Sea muddy sites, TA was very close to the overlying water concentration in Baltic sandy and





silty sites, where increases in porewater TA were not observed until depths of 10-15 cm. Resulting modelled TA flux was small and negative at the Baltic sandy and silty sites, at -4.3 and -35 µmol m$^{-2}$ hr$^{-1}$ respectively (Table 2 and Figure 4). In contrast, TA fluxes at the muddy sites were $53 \pm 95.3$ µmol m$^{-2}$ hr$^{-1}$ (Table 2). This is approximately 50% of the non-resolved

"nonriverine" TA source from Gustafsson et al., (2014), which was implied to be driven largely by denitrification, sulfur metabolism, and silicate weathering. The minor down-core differences in TA in the Baltic Sea sandy sites are due to enhanced mixing in sediments with relatively low microbial activity. An analysis (Sections 3.4-3.6), supporting the following factors as key in sustaining low or negative net TA fluxes in sandy Baltic Sea sites: 1) low net denitrification, 2) complete re-oxidation of sulfide and Fe(II) in surface sediments.

**3.3 Sediment-water fluxes of other constituents (Baltic and North Sea)**

Net sediment-water fluxes are shown together in Figure 4, with the bars representing average PROFILE-modelled fluxes, while the fluxes derived from incubation experiments and $^{224}$Ra decay balances are depicted as the colored points. Typically, PROFILE-modelled fluxes were at least 10x smaller than fluxes derived from the $^{224}$Ra decay balance. This is consistent with enhanced advective forcing that is captured by the decay-balance approach, but missed by the incubation

experiment and PROFILE model, which are in closer agreement. In addition to the sediment-water fluxes described above, the PROFILE model also generates estimates of net internal production/consumption, representing a superimposition of combined biological and abiotic sources/sinks in each model zone of the sediment (Figure S2). By convention, positive irrigation or production values indicate a net increase in concentration.





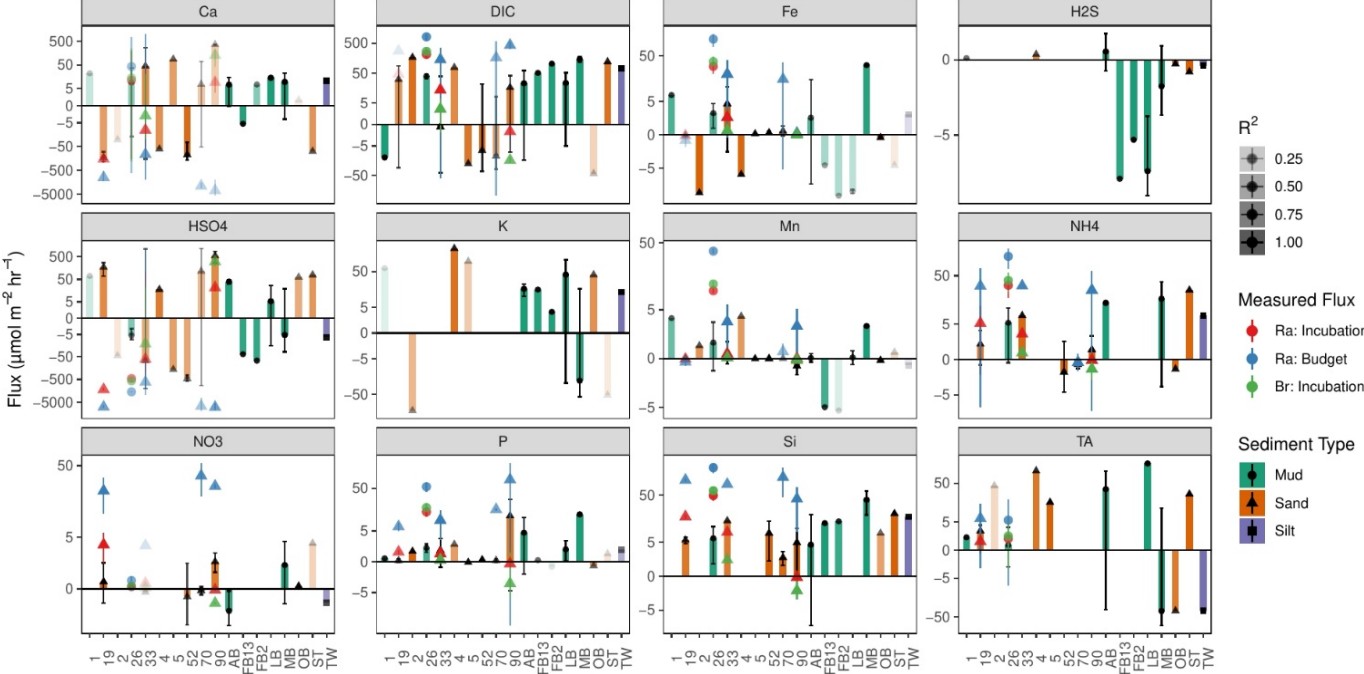

**Figure 4.** Sediment-water fluxes ($\mu$mol m$^{-2}$ hr$^{-1}$) for all sites in Baltic (sites AB – TW) and North Seas (sites 1 – 90), derived from PROFILE modelling (colored bars), with incubation and decay rates for comparison from the North Sea only (colored points). Bars with greater transparency indicate PROFILE flux results of relatively lower confidence ($R^2$). The error bars in this figure represent the standard deviation of fluxes modeled from replicate cores at each site.

Some solute fluxes exhibited similar trends across all sites in the North and Baltic Seas. For example, Si fluxes were consistently positive (out of sediments) for all sites, across all methods (Figure 4). In agreement with prior work in the region (Lipka et al, 2018; Gogina et al., 2018) ammonium (NH$_4$) release was also common and positive across all locations, except for three "sandy" sites in the North Sea (52, 70, 90). Net H$_2$S fluxes were not significantly different from zero in any North Sea sites, while PROFILE-modelled H$_2$S fluxes for three Baltic sites (FB2, FB13, and LB) were negative (into the sediment). We believe that these few large negative H$_2$S fluxes were created artificially by our use of a boundary condition (flux out of the bottom) that intersects with the zone of peak H$_2$S accumulation. In contrast, other modelled solute fluxes varied more between basins (North vs Baltic) than across sites within basins. For example, TA fluxes in the North Sea sites were close to zero (albeit with low $R^2$), while PROFILE-modelled fluxes in the Baltic were larger in magnitude and more variable (Figure 4). Baltic sites were also more often sources of Ca to the water (except AB and FB13), while Ca fluxes in the North Sea were more variable and statistically less robust.

PROFILE-modelled NO$_3$ fluxes are variable and close to zero across all sites, in contrast to prior studies in the region suggesting net NO$_3$ uptake by denitrification (Brenner et al., 2016). In fact, net NO$_3$ fluxes derived from the incubation and Ra



budget were positive (out of the sediment). This is in line with porewater $NO_3$ trends, which generally show no net depletion
in porewaters relative to surface water. While canonical denitrification is likely playing some role, the small $NO_3$ fluxes in
combination with larger $NH_4$ fluxes suggest that Dissimilatory Nitrate Reduction to Ammonium (DNRA) may be a larger sink
for $NO_3$, and that any denitrification is fed by internally produced $NO_3$ rather than allochthonous $NO_3$. Therefore, in contrast
with prior studies in the region (Burt et al., 2014; Brenner et al., 2016; Voyanova et al., 2019), we suggest a very limited
potential for net TA production via denitrification in coastal North and Baltic Sea sediments. This is in line with De Borger et
al (2021) and Pätsch et al (2018), who together indicate relatively low denitrification rates roughly balanced with nitrification,
a feature that is supported by our low or negative modelled net TA fluxes (Figure S2).

DIC flux was variable across sites, but most often positive. Fluxes parameterized by Ra budgets and incubations
agreed well with previously modelled diffusive fluxes for the region (Lipka 2017, figure 70), but our PROFILE model
generated relatively low DIC fluxes. Sites with greater DIC release were also generally larger sources of Si, $NH_4$, and P, in
line with the breakdown of algal detritus as a source for both.

### 3.4 Biogeochemical sources and sinks

### 3.4.1 Miller-Tans plots

The stable isotope composition of DIC reflects the diagenetic impact from dissolved organic matter (DOM) or methane
oxidation and interactions with carbonates (e.g., Ku et al., 1999; Meister et al., 2019; Wu et al., 2017; Liu et al., 2021). Using
measured $\delta^{13}$C-DIC data, we conducted a Miller-Tans plot analysis aimed at identifying isotopic endmembers as possible DIC
sources (Figure 5). In this graphical approach, the isotopic composition of DIC multiplied by its concentration is plotted against
the concentration, with the slope indicative of the $\delta^{13}$C-DIC source value. If the remineralization of particulate organic carbon
(POC) is the only source of DIC to porewaters, the slope of the Miller-Tans plot should be the same as the isotopic signature
of $PO^{13}C$, which in these sites ranges from -20 to -25 ‰ throughout the southern North Sea (Böttcher et al., 1998, 2000;
Pollmann et al., 2021; Serna et al., 2014), from -21 to -24 ‰ in the south-western Baltic Sea (Böttcher et al., unpublished data,
Voß and Struck, 1997). Any deviation in the y-intercept from typical $PO^{13}C$ values may result from calcium carbonate
dissolution as an important modulator of DIC and TA fluxes (Winde, 2017).

Muddy or silty sites (1, 26, AB, FB-13, FB-2, LB, MB, TW) tend to have a larger range in DIC and so allow for a more
robust graphical analysis. $\delta^{13}$C-DIC endmembers for these sites were generally much higher (heavier) in comparison with the
$PO^{13}C$ references listed above, ranging from -16 to -21 ‰ (Baltic Sea) and -6 to -26 ‰ (North Sea). Due to lower microbial
activity (Al Raei et al., 2009) and enhanced porewater exchange (de Beer et al., 2005), the sandy sites have a smaller total
range in DIC concentrations and isotope compositions, challenging the application of a Miller-Tans Plot approach. Some of
the sandy sites (2, 33, 4, OB, and ST), however, yield a similar $\delta^{13}$C-DIC endmember value when compared to the muddy sites
with values ranging between -16 and -19 ‰. Other sandy sites (19, 5, 52, 70, 90) have much heavier endmembers, and ranging
from -6 to -12 ‰. This could suggest carbonate dissolution is contributing to heavier DIC values (Ku et al., 1999; Winde et





al., 2014a,b; Wu et al., 2017). Carbonate weathering in pore waters has previously been suggested to be an important DIC and TA source (Winde et al., 2014a,b; Winde, 2017; Pätsch et al., 2018; Brenner et al., 2016). However, evidence for this is weak in our sandy North Sea sites, where porewater Ca concentrations were low relative to overlying surface water, as seen in the generally negative Ca fluxes (Figures 3 and 4), indicating no significant net carbonate dissolution. That the $\delta^{13}$C-DIC

endmember at Site 1 in the North Sea (-26 ‰) is well below the PO$^{13}$C source indicates a potential DIC input from methane oxidation. Elevated surface water methane concentrations have been previously observed near Site 1 in the region around Helgoland (Bussmann et al., 2021).

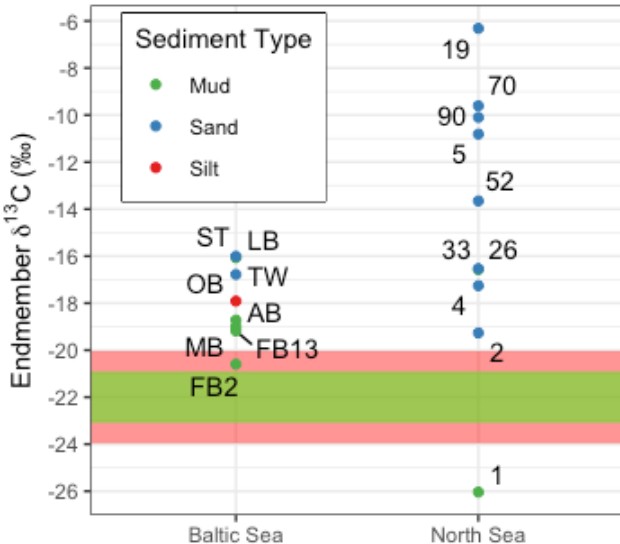

**Figure 5.** Potential end-member isotopic values for porewater DIC, derived from Miller-Tans plots of DIC * $^{13}$C-DIC vs DIC (shown in Fig S1). The colored areas represent a range in expected PO$^{13}$C values for the North (red) and Baltic (green) seas.

### 3.4.2 DIC sources and sinks: Process correlations

    While the $^{13}$C endmember indicated by the Miller-Tans plot approach above may help to identify possible electron donors, it cannot provide information as to the terminal electron acceptors that were used to respire the organic carbon

substrate. Microbial sulfate reduction (MSR) is an important mineralization process in marine sediments of this region (Jörgensen, 1989; Al-Raei et al., 2009) and leads to the production of DIC and TA (Zeebe & Wolf-Gladrow, 2001). Down-core correlations between DIC and SO$_4^{2-}$ may indicate the relative extent by which MSR is associated with DIC (and TA) production. We calculated excess DIC and sulfate as the difference between porewater and either 1) bottom water measurements, when porewater salinity, as inferred by porewater K concentration, did not change, or 2) from an empirical

relationship between water-column DIC or SO$_4^{2-}$ and K (Baltic sites FB-2 and FB-13, where salinity increased down-core).



The muddy sites in the Baltic Sea have the strongest covariation of $SO_4$ deficit and DIC, with slopes ranging from -1 to -2.6, with $r^2$ between 0.86 – 0.97, suggesting MSR is a key anaerobic pathway of DIC formation in these sites (Figure 7, green points). However, Baltic Sea sandy sites and all North Sea sites (including sites 1 and 26, which are categorized as muds) have slopes closer to zero. In the sandy North Sea sites, $SO_4^{2-}$ variation is larger than for DIC, suggesting calcification as a

possible sink for DIC. In most cases, $SO_4^{2-}$ concentrations are less than bottom water, indicating net consumption by MSR (19, 4, 5, 70, 90). However, there are also sites where some $SO_4^{2-}$ measurements were greater than bottom water (52, 33), suggesting the oxidation of $H_2S$ or $FeS_x$. This analysis is consistent with aerobic respiration as the major metabolic pathway for net DIC production in ventilated sandy sediments, likely supported by the relatively deep oxygen penetration depths.

Interestingly, North Sea mud stations (1 and 26) also have $SO_4^{2-}$:DIC slopes close to zero (-0.3 and 0.009, respectively). Therefore, MSR does not appear to be a dominant net DIC source in these muds, even though oxygen penetration depths are low and gross MSR rates are relatively high (Jørgensen, 1989). Instead, it is likely that MSR is coupled with internal sulfide oxidation. Potential sources of this sulfate in cases where it appears to be produced (North Sea sites 1, 33, and 52, and Baltic site ST) include sulfide oxidation and $FeS_x$ oxidation, both of which would be increased by intermittent deeper oxygen

penetration. The potential sink for DIC can be better understood when considering the TA:DIC ratios in the next section.

### 3.4.3 TA sources and sinks: Process correlations

We can directly assess the effect of metabolic processes on net TA production for a subset of sites where porewater TA was measured with a sufficient analytical precision. In most of the sandy sites, the down-core variation in TA was less than the analytical precision of the small-volume titrations used. For the sites where porewater TA data are available, TA and

DIC are strongly correlated, even in some sandy sites with a much smaller range in both analytes (Figure 6, orange points). Muddy sites in the Baltic Sea (AB, LB, MB, and the silty TW) have a larger range in both TA and DIC, with slopes (i.e., ratio of TA:DIC) between 0.78 and 1.095. The two sandy Baltic sites (OB and ST) have similar slopes (1.16 and 0.76 respectively), although the range in both variables is much smaller. North Sea Site 1 (muddy) has a slope of 1.12, while the sandy sites have a broad range of 0.13 (site 2) to 0.65 (site 5) and 0.8 (site 4). Modelled DIC fluxes were much larger than TA fluxes for all

North Sea sites (Figure 4; see scale), consistent with measured porewater TA:DIC slopes below 1:1 (Figure 6; orange points).

This is in part due to the likely dominance of aerobic respiration in ventilated North Sea sediments, which in contrast with anaerobic respiration, produces no TA. This dominance of aerobic respiration in permeable North Sea sediment can be attributed to the 'redox seal', which describes the situation in percolated sediment with mobile bed forms, where virtually the complete percolated sediment layer is oxygenized, whereas transport in the anoxic layer below is restricted to molecular

diffusion (Ahmerkamp et a. 2015). Since porewater advection is several orders of magnitude more efficient in the transport of substrates and products than molecular diffusion, fluxes across the sediment surface sustained by aerobic process substantially exceed those sustained by anaerobic processes. Furthermore, we observe relatively large apparent $SO_4^{2-}$ consumption below the oxic-anoxic interface (figure 6, green points), suggesting significant MSR even in sandy sediments. However, excess $SO_4^{2-}$

and DIC are not always positively correlated (figure 6), meaning that the DIC produced by MSR-driven DIC may be consumed

insitu. One potential sink for this MSR-driven DIC production might be CaCO$_3$ precipitation in the oxic zone, which is densely

colonized by calcifying infauna.

Trends between excess TA and apparent SO$_4^{2-}$ consumption are like those for excess DIC (Figure 6; purple points). Baltic muddy sites have significant ($r^2 > 0.9$) slopes ranging from -1.3 to -2.1. Baltic sandy sites have slopes closer to zero, with small excess TA values, which appear disconnected from apparent SO$_4^{2-}$ reduction. North Sea sandy sites also have

slopes closer to zero, and the muddy site (Site 1) has a much shallower slope of -0.3.

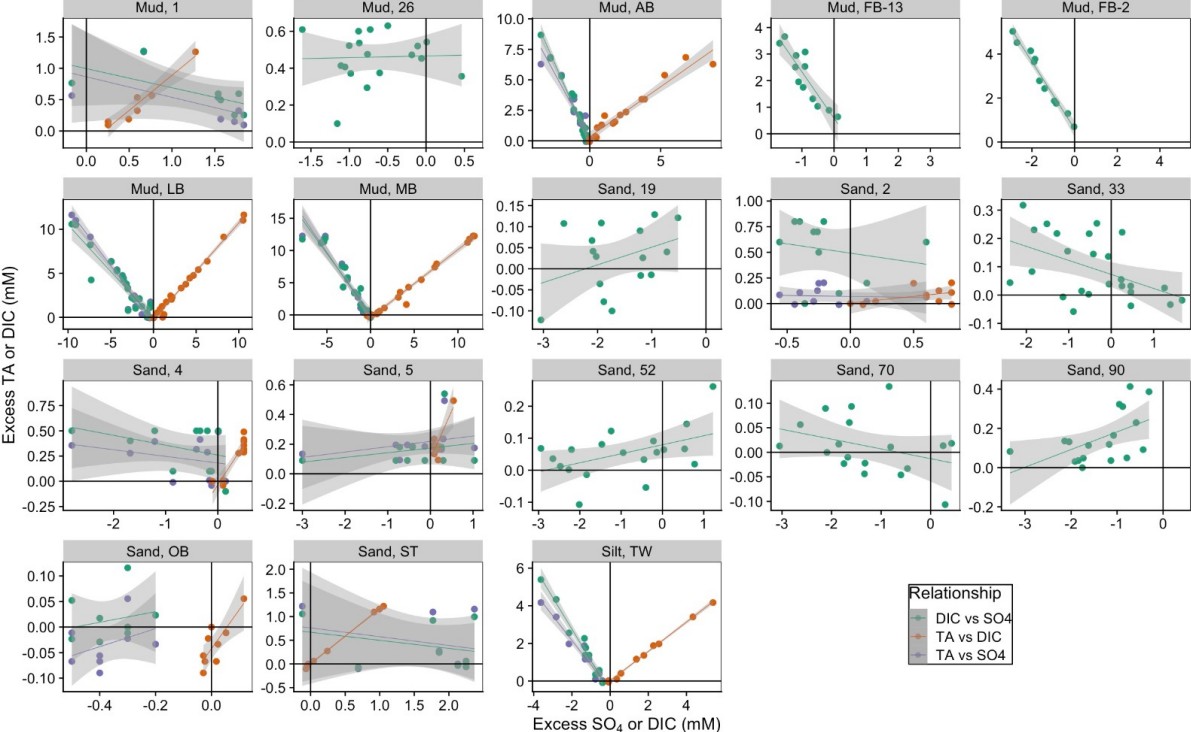

**Figure 6**. Relationships between excess DIC and excess SO$_4^{2-}$ (green points), excess TA and excess DIC (orange), and excess TA vs excess SO$_4^{2-}$ (purple). Positive values indicate a larger concentration in porewater relative to surface water.

### 3.5 PCA and regional patterns

To assess broader regional patterns in the drivers of sediment-water fluxes, we constructed a PCA (FactoMineR, Lê et. al. 2008) using a variety of selected parameters (Table 3). We included a set of biogeochemical parameters that appeared important in the above "process correlation" section, which included R$_{DIC/SO4}$ (RDS) and the ratio of P v. Fe (RFeP), although excluded relationships with TA as that parameter was not available for the entire dataset. We also included the peak concentrations of Fe, sulfide, DIC, and Si (MaxFe, MaxHS, etc) and the depth at which this was observed (DFe, DHS, etc), as



well as the first depth where sulfide was > 1 µM (DHS_1). The goal of this PCA was to identify general regional patterns in biogeochemical sources/sinks of TA and to relate them to readily available sediment characteristics.

**Table 3**. Input parameters used in the PCA analysis, and their definitions

| Parameter | Definition |
|---|---|
| Phi | Porosity |
| RDS | Ratio change in DIC (mM) to change in Sulfate (mM) |
| RFeP | Ratio P (µM) to Fe (µM). Only slopes with $r^2 > 0.7$ are provided, others are marked as zero |
| MaxFe | Maximum porewater Fe (µM) |
| DFe | Depth at which Fe maximum occurs (cm) |
| MaxHS | Maximum porewater Sulfide (µM) |
| DHS_1 | Depth where porewater Sulfide first exceeds 1 µM (up to total length of core) |
| MaxNO3 | Maximum porewater NO3 or NOx concentration (µM) |
| DNO3 | Depth at which maximum NO3 concentration occurs (cm) |
| MaxSi | Maximum porewater Si concentration (µM) |
| MaxDIC | Maximum porewater DIC concentration (mM) |
| MillerTans | slope of the Miller Tans plot |

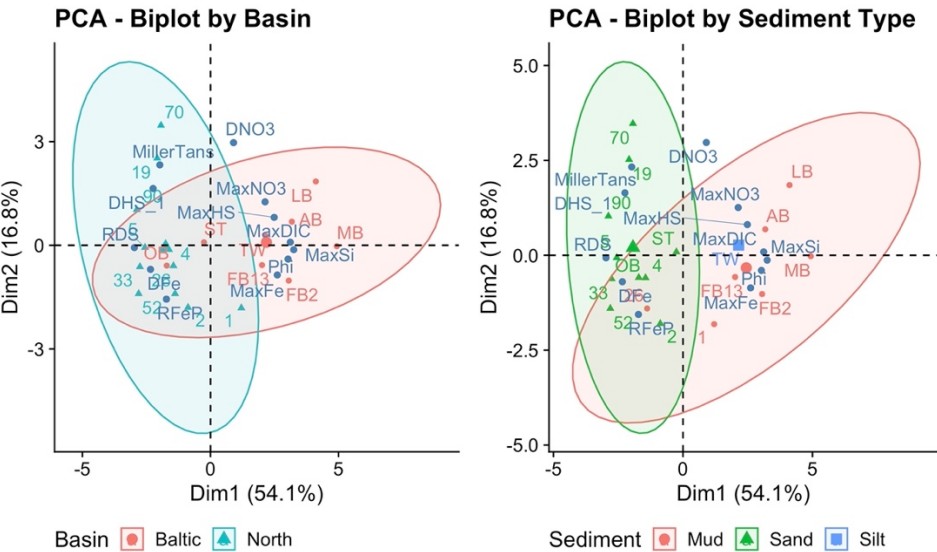

**Figure 7:** PCA Biplots separated by basin (left) and sediment type (right). Parameters (Table 3) are in dark blue, and the ellipses are calculated based on the sites, as separated by either basin (left) or sediment type (right).

Sites are strongly sorted by basin and sediment type along Dimension 1, which explains 54.8% of the variability (Figure 400 7). This is in part because the North Sea sites are dominated by sandy sites, and the Baltic sites by muddy sites. As a result, we also see strong sorting along the same Dimension 1, associated with the biogeochemical consequences of variable oxygen





penetration driven by the interaction between porosity and advective porewater exchange. These factors include: 1) porosity, as low porosities can increase oxygen penetration with disturbance from currents, waves, and storms, 2) RDS and corresponding maximum HS, which indicates the importance of MSR, an anaerobic process, 3) Depth of max Fe, which

accumulates as Fe(II) in anoxic porewaters and also is formed from the reduction of reactive Fe(III) minerals which (due to rapid (re)oxidation) accumulate at the oxic/anoxic boundaries. Additional sorting variables along Dimension 1 are related to organic matter accumulation, remineralization, and flushing of porewaters due to advection (North Sea) and include the dissolved nutrients and MaxFe. Dimension 2 explains 16.7% of the variability. Key sorting variables include the depth of Max $NO_3$, the Miller Tans slope, the accumulation of sulfide above 1 µM, and the ratio of dissolved P to Fe. For example, North

Sea sites with the highest Miller Tans slope (70, 19, 90 with potential $\delta^{13}$C-DIC sources between -6 to -11‰) are also the North Sea sites with no significant relationship between dissolved P and Fe. Together, this supports the importance of MSR in mud/silt sediments, and $O_2$-based re-oxidation of reduced sulfur species following advective exchange in sandy sites with low organic matter inputs.

## 4. Conclusion

We combined surface water and porewater $^{224}$Ra measurements to build a decay balance, which indicates that the irrigative exchange between porewater and surface water is much greater than previously thought. Implicit in this assessment is our finding that the porewater $^{224}$Ra endmember was more variable and ~twice as large as previously assumed, of importance for future Ra-based budgets, which are highly sensitive to endmember quantification (Cook et al., 2018; Garcia-Orellana et al., 2021).

In our southern North Sea sites, net sediment-water TA fluxes were very small in comparison with previous estimates in this region (Voyanova et al., 2019; Burt et al., 2014). This is also in contrast to the measured and inferred TA production rates in the Wadden Sea, which are known to be 3-5 orders of magnitude higher on an aerial basis (Santos et al., 2015), largely associated with substantial net MSR during warm seasons (Al-Raei et al., 2009). TA in ventilated North Sea surface sediments was similar to surface water concentrations, causing net sediment-water fluxes to also be very low (~10x less than for DIC).

Here, TA sources and sinks are closely balanced in North Sea sediments. We ascribe this to the potential influence of 1) DNRA / denitrification being fed internally by recycled, rather than "new" $NO_3$, 2) MSR in close balance with sulfide oxidation, and 3) carbonate mineral precipitation in the oxic zone. While positive Si fluxes could suggest silicate weathering as TA source, we cannot yet ascribe this to the erosion of biogenic (e.g., diatoms) or geogenic minerals.

   In the Baltic Sea sites, excess TA was greater in muddy regions (and close to equilibrium in silty and sandy sites), causing

modelled fluxes to be generally low and positive (muddy) or negative (sands) implying a sink in surficial sediments. At Baltic Sea sites with small positive TA fluxes (muds/silt), net TA production is likely due to net MSR (see sulfate consumption; Figure 7) and associated pyrite burial. In contrast, at sites with net modelled TA uptake (sands), modelling results indicate TA consumption in the uppermost sediment layers, which we attribute to: 1) re-oxidation of sulfide and Fe(II), 2) minimal net denitrification, as in North Sea sites, and 3) carbonate mineral dissolution.



Our findings suggest that coastal sediments of the Southern North Sea and Western Baltic Sea are currently not major sources of TA to the water column. This is because TA production in deeper anaerobic sediments is counter-balanced by re-oxidation and TA consumption in overlying oxic sediment, causing net TA fluxes to be small and variable. The seasonality of these TA fluxes and their impact on water-column carbonate chemistry and ultimately air-water $CO_2$ exchange was not assessed in this study but should be a topic of future research.

**Data Availability**


Upon publication, all data used in the preparation of this manuscript will be made publicly available at the Helmholtz Coastal Data Center (HCDC), an open-access repository, according to the FAIR principles.

**Author Contribution**

Contributor roles according to the CRediT Taxonomy

*Conceptualization*: BVD, HT, MEB
*Formal Analysis*: BVD, MAZ, AN
*Investigation, inclusive of field and laboratory work*: BVD, NL, MAZ, AN, ML, DP
*Project Administration, Resources, Supervision*: HT, MEB, DP
*Writing – original draft preparation*: BVD, MAZ
*Writing – review and editing*: BVD, NL, MAZ, AN, DP, HT, MEB

**Competing interests**

The authors declare that they have no conflict of interest.

**Acknowledgements**

The investigations were and are supported by German BMBF projects DAM-MGF, CARBOSTORE, and KÜNO SECOS-I/-
II (03F0666 and 03F0738 A–C), as well as German Academic Exchange Service (DAAD) grant #57429828 "The Ocean's Alkalinity: Connecting geological and metabolic processes and time-scales", under the BMBF "Make our Planet Great Again – German Research Initiative". Further support was provided by Helmholtz-Zentrum Hereon and the Leibniz Institute for Baltic Sea Research (IOW). We acknowledge the help of Dennis Bunke, Christian Burmeister, Florian Cordes, Andreas Frahm, Michael Glockzin, Axel Kitte, Anne Köhler, Gerhard Lehnert, Tobias Marquardt, Céline Naderipour, Sascha Plewe, Ines
Scherff, Iris Schmiedinger, Mona Norbisrath, Bettina Rust, Leon Schmidt, Justus van Beusekom, and Tristan Zimmermann during field sampling and laboratory analysis. We also thank Peter Berg for the helpful discussions about modelling with



PROFILE. The authors further wish to thank the captains and crews of R/V Heincke, R/V Elisabeth Mann Borgese, R/V Poseidon, R/V Alkor, and R/V Maria S. Merian.

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
