# Peer review of "Benthic Alkalinity fluxes from coastal sediments of the Baltic and North Seas: Comparing approaches and identifying knowledge gaps"

_EGUsphere, 2022_

## Author Comment (AC1)

EGUsphere, referee comment RC1 https://doi.org/10.5194/egusphere-2022-161-RC1, 2022 © Author(s) 2022. This work is distributed under
the Creative Commons Attribution 4.0 License.

**Comment on egusphere-2022-161**

Xinping Hu (Referee)

Referee comment on "Benthic Alkalinity fluxes from coastal sediments of the Baltic and North Seas: Comparing approaches and identifying knowledge gaps" by Bryce Van Dam et al., EGUsphere, https://doi.org/10.5194/egusphere-2022-161-RC1, 2022

Van Dam et al used three independent approaches, including Ra-224 decay balance, core incubation, and porewater profile fitting, to calculate/estimate benthic alkalinity fluxes in the sediment of both the southern North Sea and the western Baltic Sea. Part of the data used in this study have been published elsewhere. The authors also explored porewater stable carbon isotopes as well as the relationships between various parameters (alkalinity, DIC, excess $SO_4^{2-}$) for possible reaction mechanisms, for example, likely carbon source to porewater DIC, processes responsible for DIC/alkalinity changes. The overall conclusion is that benthic alkalinity fluxes in the studied regions are substantially smaller compared to the results obtained from prior studies in these areas, even though the estimates do vary because of the different approaches taken in this work.

The manuscript is largely well written, and the authors have done a good job tying together both historical and more recent collected data and applying the three techniques to examine fluxes. The detailed geochemical analyses, for example the interpretation of porewater stable isotopes and apparent reaction stoichiometry, are very informative. That being said, this work on one hand lacks some details on the methodology in the geochemical analysis and modeling, for example not all reported porewater constituents have corresponding analytical methods ($NO_3^-$, $K^+$ etc) and none of the methods has precision information, and the parameterization of the PROFILE model seems to offer no context regarding where these values are from; it also seems to bog down in details of flux values of many constituents coming from different methods while there is little quantitative understanding of indeed how much benthic alkalinity is exported to the water column on a regional scale, other than the fact that the values are much smaller than thought. It may be of interest to readers to show the flux ratios of constitutes that could be illuminating for understanding overall reaction stoichiometry (e.g., carbonate dissolution/precipitation) based on the PROFILE model calculations, and perhaps complement the discussion with both the stable isotopes and porewater ratio information, so the latter two do not necessarily stand alone. In the end, the authors stated that seasonality of this flux needs to be researched, among other things. However, given the fact that data from the four cruises already spanned different seasons, it is unclear why seasonality cannot be addressed here, or at least some effort can be taken in this work.

The PCA analysis is interesting, although it also provides little quantitative knowledge on understanding benthic fluxes other than showing that both study region and sediment particle size matter for benthic fluxes, which is not surprising but hardly unexpected. The choice of the input parameters also seems arbitrary and more contextual information is needed if the authors decided to keep this section.

In figure presentations, the authors almost exclusively used bar charts, and some of the figures (Fig. 4) uses fairly complex notation schemes. It will serve readers better if the

authors could consider using correlation plots as an additional visual aid to compare and contrast values of the same nature but obtained from different means.

We appreciate the detailed and critical review offered by both referees. In response to their collective suggestions, we have made major revisions to the submitted manuscript. This involved creating new versions of Figures 1-5, removing section 3.5 ("PCA and regional patterns"), adding information to the methods descriptions, and revising the discussion and conclusion (section 3) for improved clarity. We feel that these changes were very helpful, as the manuscript now more clearly conveys our methods and key findings.

The first Referee also suggests above that we use ratios of PROFILE-modelled fluxes to further investigate net reaction stoichiometry. Unfortunately, though, the only ratios with correlations of $R^2 > 0.5$ were DIC:NH4 and DIC:Si. Both NH4 and Si exhibited slopes of ~5:1, consistent with the remineralization of detrital phytoplankton biomass. This is consistent with our characterization that net DIC production was relatively low, due to closely balanced redox and precipitation-dissolution cycling, with any residual flux related to the breakdown of organic matter (last sentences of section 3.3). For this reason we are hesitant to add an additional figure and text to the manuscript showing these flux ratios, but if either reviewer feels that this would be a useful addition, we would be happy to do so in a subsequent revision.

Below are some detailed comments:

Fig. 1 add coordinates axes to the map.

The map has been updated with coordinate axes and an improved bathymetric representation.

L112, spell out IOW even though it appears in the affiliations already?

The full name for IOW is now given here as well as in the affiliations.

L113, is it HCl too?

Yes, this is now described.

L133, "at IOW" appears twice.

The extra wording was removed

Section 2.2, please list the precision for all constituents analyzed, even if they may have appeared elsewhere for example prior publications. Later in the text, for example Fig. 4, it seems that not all solutes are mentioned in this section.

As some parameters were measured by different labs for different cruises, a description in the text would be quite lengthy and cumbersome. So, instead we have added precision information for all analytes discussed in the main text to the supplemental table S1.

Section 2.4—2.6 seem to be more appropriate as subsections of 2.3 (2.3.1, 2.3.2, and 2.3.3) because 2.3 lays out all three techniques but sections 2.4-2.6 elaborate them.

Thank you for the suggestion, and I agree, sections 2.4-2.6 are now sub-sections of 2.3

L179, only DIC and Ra were measured? In fact, sections 2.4-2.6 lack general information on what were collected and modelled. Even though the lab analysis section (2.2) mentioned analytical methods for porewater parameters, it is unclear whether all or parts of the parameters were used for all incubations/modeling studies. For Table 1, are these values part of the input? If so, how were the values obtained?

No, there were a few other parameters measured (nutrients) in the incubations, but these data were not available in time for the preparation of this manuscript. This is why most of the fluxes we describe were modelled in PROFILE, which used a porewater dataset that was more complete across cruises.

Regarding the PROFILE model parameterization, we have added a sentence to section 2.3.3 explaining that Biodiffusivity and irrigation coefficients were arrived at by an optimization exercise. The statistical parameters (max deviation and level of significance) were chosen following a personal communication with the developer of the model, Peter Berg (as mentioned in the acknowledgement section).

Fig. 2, there is no discussions on Ra-223 throughout the text, where does this information come from? In figure caption please note these sites are from the North Sea.

Good point, I have removed the panel with the 223Ra-derived irrigation rates, and added text in the caption to clarify that this is for the North Sea only (this is also explained in the main text).

L236, if TA values are reported to the second decimal place, it would imply that the precision only reached 0.1 mM or 100 μM at best as by analytical chemistry convention the last digit is used as an estimate, then the bottom-pore water TA difference of 2-4 μM appears unrealistic, please clarify.

The analytical precision of our TA titrations (~2%) is indeed above these small bottom-pore water TA differences of 2-4 μM, and is the reason that our TA fluxes are mostly close to zero.

L237, briefly state the method that Voynova et al. (2019) used to inform readers.

Yes, good point, Voynova et al 2019 did not directly measure TA flux, but rather estimated it based on their observation of a seasonal increase in water-column TA throughout the southern north sea. This is now briefly described in the main text.

L240-241, in Table 2, TA flux at the maximum 33.6 μmol/m2/hr, but the statement that two prior studies reported results "more in line" values is confusing. Please clarify as these values are nowhere close to what's reported in this section.

Yes, we can see how this is confusing. While these two studies (Pätsch et al., 2018 and Brenner et al., 2016) reported low net TA fluxes relative to other work, their results were still above ours. The text has been revised to clarify this point.

"237.5-275" should be either 237.5-275.0 or 238-275. Significant figures matter.

Thank you, this has been corrected. We converted these values from the units presented in Brenner et al., 2016.

L248-249 and L254, are they the same thing? If so, merge to reduce redundancy.

No, these are not the same. The first reference is simply pointing out the site with the largest modelled TA fluxes, while the second statement is about TA fluxes across all Baltic muddy sites. Thank you for bringing up this point, and we have tried to clarify the statements in the revisions.

L291, the larger NH4+ flux may have organic matter breakdown component as well, see L300. If DNRA is an important process, some references to back it up would be helpful.

This is a good point. Our approach is certainly not capable of resolving these questions related to the processing of internal vs "new" N. Prior work in the region has shown that DNRA is present, albeit at relatively low rates, so we are not able to distinguish NH4 derived from DNRA and that from simple OM degradation. We have added text to clarify this point, along with additional references to support our broader claim of limited net denitrification, which is fed largely by nitrification.

Fig. 5, it seems that the site label and data points are misaligned so it's difficult to see where some data points are from.

Thank you for the suggestion. I have replaced Figure 5 with a new version where the labels are moved to avoid overlapping.

L343-344 is repeated in. L350.

Thank you, the earlier statement has been removed

L350 paragraph, Site 1 is said to have methane as the possible organic carbon source, here the authors suggested that shallow O2 penetration and high MSR together with sulfur recycling does not lead to net sulfate reduction. As this is a "mud" site, the interpretation seems counterintuitive as marine sediments of this nature in general would see a reduction in sulfate concentration (high MSR rate and low permeability).

Regarding methane, this is a very interesting point, and one that I feel calls for its own dedicated study, but unfortunately, this is again a case where our study comes up short on robust explanations. Recent work at a nearby site (between our sites 1 and 52) did find quite high methane concentrations in shallow porewater (Aromokeye et al, 2020), but another study very close to our site 1 showed a SMTZ well below even the deepest samples that we collected. We interpret the endmember 13C-DIC at site 1 as indicative of depleted C following methane oxidation, although the ultimate source of this methane is uncertain. This is consistent with the work of Krämer et al., 2017, (conducted at the same general location near Helgoland), which found elevated methane concentrations below 100cm, and a SMTZ between ~20-100 cm depth. The short length of our core at this site (~20cm) means that our samples were all above the SMTZ, and we therefore expect that all methane should have already been oxidized by available TEAs (SO4, NO3, Fe, etc).

We should also make clear that gross sulfate reduction is likely relatively high here, but balanced closely by re-oxidation. We understand that there can be confusion when gross and net rates are discussed together, and have revised the text to clarify this point.

Aromokeye, D. A., Kulkarni, A. C., Elvert, M., Wegener, G., Henkel, S., Coffinet, S., Eickhorst, T., Oni, O. E., Richter-Heitmann, T., Schnakenberg, A., Taubner, H., Wunder, L., Yin, X., Zhu, Q., Hinrichs, K. U., Kasten, S. and Friedrich, M. W.: Rates and Microbial Players of Iron-Driven Anaerobic Oxidation of Methane in Methanic Marine Sediments, Front. Microbiol., 10(January), 1–19, doi:10.3389/fmicb.2019.03041, 2020.

Krämer, K., Holler, P., Herbst, G., Bratek, A., Ahmerkamp, S., Neumann, A., Bartholomä, A., Van Beusekom, J. E. E., Holtappels, M. and Winter, C.: Abrupt emergence of a large pockmark field in the German Bight, southeastern North Sea, Sci. Rep., 7(1), 1–8, doi:10.1038/s41598-017-05536-1, 2017.

---

## Author Comment (AC2)

EGUsphere, referee comment RC2 https://doi.org/10.5194/egusphere-2022-161-RC2, 2022 © Author(s) 2022. This work is distributed under
the Creative Commons Attribution 4.0 License.

**Comment on egusphere-2022-161**

Anonymous Referee #2

Referee comment on "Benthic Alkalinity fluxes from coastal sediments of the Baltic and North Seas: Comparing approaches and identifying knowledge gaps" by Bryce Van Dam et al., EGUsphere, https://doi.org/10.5194/egusphere-2022-161-RC2, 2022

The authors measured alkalinity fluxes and other related geochemical parameters in North Sea and Baltic Sea sediments. A key strength of the study was the use of a wide variety of approaches to estimate alkalinity fluxes. The work is interesting and topical given the possible role of alkalinity production in mediating $CO_2$ uptake in the coastal ocean. Overall, although the text was generally well written, this work felt like a rough draft rather than a polished manuscript ready for submission. The tables and figures were generally poor quality in terms of their visual appeal and ease of interpretation. The methods were incompletely described and the results and discussion unfocused.

We appreciate the detailed and critical review offered by both referees. In response to their collective suggestions, we have made major revisions to the submitted manuscript. This involved creating new versions of Figures 1-5, removing section 3.5 ("PCA and regional patterns"), adding information to the methods descriptions, and revising the discussion and conclusion (section 3) for improved clarity. We feel that these changes were very helpful, as the manuscript now more clearly conveys our methods and key findings.

Specific comments

Ship board incubations – I don't understand why fluxes of DO, TA and DIC (and nutrients) were not measured in these incubations? This is probably one of the most common approaches (along with chambers) for measuring fluxes.

Thank you for the comment, and I agree that it would have been nice to have TA and DIC measurements in the flux incubations. But, unfortunately the assessment of alkalinity and carbon fluxes was not an objective of the original field work. We do have paired O2 and DIC fluxes for a few sites in the North Sea, from the HE541 cruise, which are presented below. The solid and dashed lines are the 1:1 and 2:1 reference lines respectively. We chose not to include this figure in the submitted manuscript because of the limited spatial coverage (just 3 sites).

[Figure]

Methods what was the precision of the TA analysis and all other methods?

This information has now been added to the SI in table S1.

I don't think the fluxes presented for Fe, Mn, Ca, H2S, K and HSO4 (SO42-) were meaningful as these solutes either oxidise (H2) and precipitate (Fe, Mn), or the small concentration differences between the sediment and the water column are probably random (especially without information on precision).

In an effort to simplify the figures, the elements not discussed in the text have been removed from figures 3 and 4.

Figure 2 and others. Label the x axis!

Yes, thank you for this reminder, all figures have been updated with x-axis labels

Figures 3 and 4 are a bit overwhelming and hard to interpret. Can the authors find a way to present the data more clearly (this will be easier when the analytes noted above are dropped).

We agree that figures 3 and 4 were overwhelming, and have reduced the number of elements considered so as to focus on those that are important for our discussion.

Results and Discussion

I would suggest that results and discussion be separated. This will allow a more focused discussion on the key points of interest. At the moment there is a lot of focus on details and jumping across different ideas. What are the key factors controlling alkalinity production based on your data set? It might be helpful to separate muds and sands into different sections.

We appreciate the helpful suggestion, and have put effort into revising this section for clarity, which indeed was difficult to follow in places. Following this, and the removal of the PCA section, we feel that the results/discussion is now much improved and can stay as a combined section.

I don't think the PCA plot helped us understand the geochemistry here. This approach is useful when the a-priori mechanistic links between variables is unclear. I think the links between the geochemical variables here are well known and understood and the interpretation of the PCA plots just re-iterates this understanding.

The PCA plots and discussion surrounding this was removed, following the recommendation of both Reviewers 1 and 2.

The miller-tans plots suggest carbonate dissolution is important, particularly in the North Sea sands. It is noted this contradicts low porewater Ca concentrations, but I doubt if the method has sufficient precision to really make this assessment. Also, it is likely dissolution and precipitation are occurring simultaneously?

Indeed, the carbonate dissolution that we infer is likely matched by re-precipitation, either in-situ or in overlying sediment layers. This is supported by our very low (and variable) net Ca fluxes, which do not indicate any appreciable net dissolution of carbonate material (as described in section 3.4.1).

Conclusion

Pyrite burial is suddenly mentioned as a factor in alkalinity production with no prior mention in results or discussion.

Yes, thank you for pointing this out. We did not measure or estimate pyrite accumulation in any way, so I have removed this statement from the conclusion.

---

## Author Response (AR2)

We appreciate the additional comments and suggestions provided by the reviewers, and have accordingly made minor revisions to the manuscript.